# Botulinum Neurotoxins beyond Neurons: Interplay with Glial Cells

**DOI:** 10.3390/toxins14100704

**Published:** 2022-10-13

**Authors:** Siro Luvisetto

**Affiliations:** Institute of Biochemistry and Cell Biology, National Research Council of Italy, Via Ercole Ramarini 32, Monterotondo Scalo, 00015 Rome, Italy; siro.luvisetto@cnr.it

**Keywords:** botulinum, glia, peripheral nervous system, central nervous system, animal models

## Abstract

In recent years, numerous studies have highlighted the significant use of botulinum neurotoxins (BoNTs) in the human therapy of various motor and autonomic disorders. The therapeutic action is exerted with the selective cleavage of specific sites of the SNARE’s protein complex, which plays a key role in the vesicular neuroexocytosis which is responsible for neural transmission. The primary target of the BoNTs’ action is the peripheral neuromuscular junction (NMJ), where, by blocking cholinergic neurons releasing acetylcholine (ACh), they interfere with neural transmission. A great deal of experimental evidence has demonstrated that BoNTs are also effective in blocking the release of other neurotransmitters or neuromodulators, such as glutamate, substance-P, and CGRP, and they can interfere with the function of glial cells, both at the peripheral and central level. The purpose of this review is to provide an update on the available experimental data from animal models that suggest or confirm the direct interactions between BoNTs and glial cells. From the data collected, it appears evident that, through mechanisms that are not yet fully understood, BoNTs can block the activation of spinal glial cells and their subsequent release of pro-inflammatory factors. BoNTs are also able to promote peripheral regeneration processes after nerve injury by stimulating the proliferation of Schwann cells. The data will be discussed in consideration of the possible therapeutic implications of the use of BoNTs on those pathological conditions where the contribution of glial cell activation is fundamental, such as in peripheral and central neuropathies.

## 1. Introduction

Botulinum neurotoxins (BoNTs) are produced by clostridium botulinum in the subtypes of seven serotypes, named from A to G [1,2], in addition to the recently characterized serotype X, and a certain number of chimeric neurotoxins [3,4]. In this paper, the different types of BoNTs will be reported using the acronym BoNT/Y, where Y stands for serotypes A to X. If the subtype is not specified, then the acronym BoNTs will be used in a generic manner; where necessary, the commercial name will be used.

All BoNTs have similar structure, consisting of two chains (L-chain, 50 kDa, and H-chain, 100 kDa) that are linked by a single disulfide bridge, with the L-chain being the catalytic domain and the H-chain being the receptor-binding and translocation domain [5]. BoNTs act as powerful blockers of synaptic vesicle fusion at the peripheral neuromuscular junction (NMJ), where they block the release of acetylcholine (ACh). Inhibition of vesicular ACh release is achieved via the cleavage of SNARE proteins, which constitute the protein complex essential for the vesicular release evoked by the action potential at the NMJ. Different BoNT serotypes cleave different SNARE proteins: BoNT/B, D, F, G, and X cleave VAMP at single and different sites; BoNT/C cleaves both syntaxin and SNAP-25, whereas BoNT/A and E cleave SNAP-25 at different sites [6]. 

In recent years, many basic scientific studies, and a great deal of clinical evidence, demonstrated that, apart from the canonical anticholinergic effects at the NMJ, BoNTs are effective in inhibiting the ACh release at sites other than the NMJ, in addition to inhibiting the release of neurotransmitters other than ACh [7]. Currently, there is no doubt that BoNTs can block the release of excitatory neurotransmitters and neuropeptides, such as glutamate, substance-P, and CGRP. These substances, along with certain actions, are strongly involved in pain modulation [8]. Moreover, the finding that BoNTs may block their release provided studies, that were conducted on both animal models and in clinics [9,10], with the impetus to suggest the use of BoNTs as an analgesic for chronic pain conditions that did not respond to other analgesic drugs; therefore, BoNTs have been suggested as a third line analgesic treatment [11].

Many studies have provided evidence for the fact that, apart from the canonical action on neurons, BoNTs may interact also with glial cells, both in the central (CNS) and peripheral (PNS) nervous system. Historically, the first evidence for an interaction between BoNTs and glial cells, came from a series of in vitro studies in cultured astrocytes. Jeftinija et al. [12] demonstrated that the pre-treatment of cultured astrocytes with BoNT/A or BoNT/C decreased both the baseline and the bradykinin-evoked release of glutamate. Verderio et al. [13] demonstrated that BoNT/B and BoNT/F are internalized to culture astrocytes, whereas Araque et al. [14] demonstrated that a microinjection of cultured astrocytes with light chain BoNT/B strongly reduced SNARE protein-dependent glutamate releases. Moreover, it was demonstrated that BoNT/A also reduced the extracellular high K^+^-induced increase in glutamate that was released from the astrocytes [15]. Other in vitro studies showed that BoNT/A blocked the uridine triphosphate-stimulated ATP release from both cultured astrocytes that were isolated from rat cortexes [16] and Schwann cells (SCs) that were cultured from the sciatic nerve [17]. An additional effect of BoNT/A on cultured SCs was found by Marinelli et al. [18]; in ex vivo experiments, Marinelli et al. [18] found that, after a peripheral injection of BoNT/A in mice that were subjected to a ligature of the sciatic nerve, cleaved SNAP-25 co-localized with astrocytes. This finding has been considered as strong evidence for the possible transcytosis of BoNT/A from neuronal cells to astrocytes. Blocking the release of astrocytic glutamate from BoNT/A may contribute to the reduction of pain. 

As the aim of this review is to give a description of the current findings on the interaction between BoNTs and glial cells, further evidence will be discussed below. Given that there may be a possible therapeutic application of BoNTs, regarding pathologies in which glial cells are deeply involved in the onset and maintenance of the disease, particular emphasis will be given to the evidence concerning the direct effects of BoNTs on glial activation for the treatment of neuropathic conditions and/or neuronal lesions. Finally, given the fact that the last review on this subject dates back to 2018 [19], this descriptive review aims to update the current literature, starting from 2018, with the aim to stimulate future studies on this important topic.

## 2. Search Strategy and Inclusion Criteria

The online search was conducted on the databases PubMed/MEDLINE, ScienceDirect, Scopus, and Web of Science, and it included articles from January 2018 up to September 2022. The search on the databases were performed using all paired combinations of the term ‘botulinum’ with (AND) one of the following terms: ‘glia’, ‘glial’, ‘microglia’, ‘macroglia’, ‘astroglia’, ‘astrocytes’, ‘oligodendrocyte’, ‘Schwann’, ‘satellite glia’, ‘ependymal’, ‘radial glia’, and ‘enteric glia’. The experimental articles included both animal-based and clinical studies; the only studies which were excluded were those that were conducted with botulinum C3, a toxin which secretes the enzyme C3, that selectively causes the mono-ADP-ribosylation of the small GTPases Rho-A, -B, and -C, thus inhibiting cell–Rho signaling. Such exclusions are justified given the fact that the mechanism of action for the botulinum toxin C3 is completely different from that of classical BoNTs, which act on SNARE proteins; this is the focus of the present review.

## 3. Glia

Glia are non-neuronal cells existing both in CNS and PNS (Figure 1). The main difference between glia and neurons, is that glial cells do not produce action potentials, and thus, for many years, their function was considered to be exclusively structural, providing only functional and protective support for neurons. For this reason, they were named “glia”, which is a Greek translation for “glue”. The “gluing” function of glia is now considered to be minor, and it has been widely recognized that glial cells are actively involved in the modulation of the neuronal environment, and the regulation of many neuronal functions, such as the nerve firing discharge, brain plasticity, the immune and inflammatory response, the formation of the myelin sheath of axons, and recovery from nerve injuries [20]. Different types of glial cells play different specific roles.

Glial cell types belong to two main categories: macroglia and microglia. Macroglia are involved in the regulation and optimization of neuronal function, whereas microglia have phagocytic properties and help make the neuronal environment safer. Macroglia exist in four main forms in the CNS, namely, the oligodendrocytes, astrocytes, ependymal cells, and radial glia. In the PNS, they exist in three main forms, namely, the Schwann cells, satellite glial cells, and enteric glia. 

Oligodendrocytes are cells that line axons in the CNS with their cell membrane, which forms the myelin sheath, thus allowing electrical signals to propagate along axons more efficiently [21,22]. A single oligodendrocyte provides insulation for multiple neurons. Oligodendrocytes also support the metabolic needs of the axons in nerve cells [23]. 

Ependymal cells form layers that delimit the space in brain ventricles and the spinal cord with a continuous sheet of epithelium [24,25]. The main function of ependymal cells is the production of cerebrospinal fluid [26]. Since they are ciliated cells, they help to distribute neurotransmitters and hormones, and they also contribute to osmotic control within the brain via the regulation of glucose and the uptake of ions [24]. 

Radial glia comprise a subgroup of glial cells that include Bergmann and Muller cells. Radial glia are only found in specific areas of the CNS [27], namely, the cerebellum (Bergmann cells) and retina (Muller cells), where they modulate neurotransmission and optimize how information is processed. During brain development, radial glia also function as neuronal stem cells [28].

Astrocytes, also called astroglia, are the most abundant type of macroglia in the CNS [29]. They regulate the external chemical environment of neurons by recycling the neurotransmitters that are released during synaptic transmission [30]. Astrocytes are also involved in vasoconstriction and vasodilation, which are actions that occur after substances such as arachidonic acid have been produced, as their metabolites are vasoactive. They also connect neurons to blood vessels and help to maintain the permeability of the blood–brain barrier, where they sense the levels of glucose and ions and regulate their flow into, or out of, the brain [31]. 

Schwann cells (SCs) in the PNS operate similarly to the oligodendrocytes in the CNS. They provide myelination to nerve axons and they modulate the extracellular fluid [32]. Unlike oligodendrocytes in the CNS, where a single oligodendrocyte myelinates multiple axons, a single axon in the PNS is myelinated by multiple SCs. SCs not only have a myelinating function, but they also exhibit phagocytic activity. In fact, together with the infiltrating macrophages, they help to clear cellular debris after neuronal lesions, thus they favor the regrowth of PNS neurons [33]. 

Satellite glial cells (SGCs) are small, flattened cells found in the PNS, where they surround sensory and autonomic ganglia. These cells, which help to regulate the external chemical environment, are highly sensitive to injury and inflammation, and they appear to contribute to pathological states, such as acute and chronic pain [34]. In some respects, their functions are similar to the astrocytes in the CNS [35,36]. In sensory neurons, SGCs regulate K^+^ levels, and the neurons respond to the evoked potentials [37]. 

Enteric glial cells are found in the lining of the intestines, among the enteric ganglia in the digestive system, in smooth muscle layers, and in gut mucosa. They have many roles that are related to homeostasis and the muscular digestive processes in the enteric nervous system [38]. They are implicated in peristalsis, and they encourage contact between the different cells of the intestinal wall [39]. Enteric glia are also involved in the pathogenesis of inflammatory diseases concerning the enteric nervous system, such as inflammatory bowel diseases, and of functional gastrointestinal diseases, such as irritable bowel syndrome [40].

Microglia are found in all regions of the brain and spinal cord. Microglia are specialized macrophages that are capable of phagocytosis, which eliminates cellular debris and toxins, thus protecting CNS neurons [41]. Microglia are small, relative to macroglia, with changing shapes and oblong nuclei [42,43]. As macrophages, microglia hold fundamental roles in terms of nervous tissue immunity and inflammatory responses [44]. They are also implicated in many other processes that are involved in the optimization of different brain circuits which enable cognitive development. In fact, microglia perform their “cleaning action” by phagocyting previously formed synapses that are no longer useful [45].

## 4. Interactions between BoNTs, Microglia, and Astrocytes

The main evidence for the interactions between BoNTs, astrocytes, and microglia comes from studies on pathological pain in animal models. Pathological pain is characterized by an amplified response to normally harmless stimuli and an amplified response to acute pain. With conditions that cause pathological pain, which results from a dysfunction in the neuronal activity of sensory neurons, the activation of spinal glial cells, microglia, and astrocytes, contributes to the development and maintenance of chronic pain [46,47,48,49]. Glial cells are activated by the neuronal release of neuromediators, including substance P, glutamate, and fractalkine. The activated glia may release other mediators that, via a feedback action on glia and neurons, produce an amplification of the pain signals. Critical mediators that sustain the amplification of pain have been demonstrated to be pro-inflammatory cytokines [50,51]. In a previous review, Rojewska et al. [19] provided a detailed analysis of the evidence that shows a modulatory interaction between BoNT/A and microglia, astrocytes, and neurons under neuropathic pain conditions; they had a particular interest in clarifying how BoNT/A may affect spinal neuron–glial interactions. Starting with the review of Rojewska et al. [19], which gives a comprehensive review on these topics before 2018, the current chapter of this review aims to provide an update on the effects of BoNTs on microglia and/or astrocytes by using the findings of newer studies.

Before analyzing the interaction between BoNTs and microglia in detail, some preliminary fixed points must be addressed. It is well recognized that, under pathophysiological conditions, microglia can act in two different forms: a classic pro-inflammatory phenotype and an alternative anti-inflammatory phenotype [52,53]. As a pro-inflammatory phenotype, microglia release TNF-α, IL-1β, IL-12, IL-23, and pro-inflammatory cytokines, which exacerbate inflammation and tissue injury. In contrast, as an anti-inflammatory phenotype, they release TGF-β, IL-4, IL-10, Il-13, VEGF, BDNF, PDGF, anti-inflammatory cytokines, and growth factors, which suppress inflammation and promote tissue recovery, respectively [47]. Differentiation of microglia towards the pro-inflammatory phenotype often occurs during neuropathic pain, and a transition from a pro-inflammatory to an anti-inflammatory phenotype may represent an innovative therapeutic strategy for relieving neuropathic pain [19,47]. 

In a neuropathic pain model that involved a chronic constrictive injury (CCI) to the sciatic nerve of a rat, Gui et al. [54] found that the subcutaneous injection of BoNT/A (10–20 U/kg Botox^®^ into the metatarsal surface; three days after CCI) promoted the polarization of a microglial to become an anti-inflammatory phenotype. This finding correlated with the decreased expression of the microglial purinergic P2X7 receptor, along with the increased mechanical withdrawal threshold and thermal withdrawal latency. The reduced expression of P2X7 receptors was also confirmed by in vitro assays performed in a microglial cell line stimulated by lipopolysaccharide (LPS). The exact mechanism by which BoNT/A reduced the expression of P2X7 receptors, and thus, the mechanism that caused the microglia to become an anti-inflammatory phenotype, remains unknown. In another report from the same laboratory [55], the authors confirmed that BoNT/A attenuated CCI-induced neuropathic pain in rats by slowing the release of pro-inflammatory factors from activated microglia, as well as mitigating the expression of SNAP-23. Reducing the SNAP-23 expression in microglia occurs by targeting toll-like receptor 2 (TLR2), and its adaptor protein, MyD88. Toll-like receptors are normally expressed in immune and glial cells, where they regulate innate and adaptive immunity. 

It is worth considering that the Botox dose used in references [54,55] appear particularly high. In fact, the dose of 20 U/Kg would represent a dose of 1500 units for a 75 Kg adult, which is exceptionally high (15 vials of 100 U per vial). In light of this, translating the results in references [54,55] from rats to humans seems questionable; however, it should be also noted that doses of BoNT/A in animal models cannot be simply converted into therapeutic doses for humans on the basis of weight ratios, rather, they must be appropriately chosen on the basis of toxicity criteria.

The role of TLR2-mediated neuroinflammation was also evidenced by Chen et al. [56], who demonstrated that the unilateral subcutaneous facial injection of BoNT/A (0.18 U of Lanzhou manufactured BoNT/A into the whisker pad), in a trigeminal neuralgia model induced by CCI of the distal infraorbital nerve in mice, attenuated bilateral trigeminal neuropathic pain behaviors and inhibited the upregulation of microglia in TLR2 expression. 

Altogether, these results are confirmed by the findings of Piotroska et al. [57], who revealed that BoNT/A inhibits the expression of pro-inflammatory factors through the modulation of NF-kB, p38, and ERK1/2. Moreover, it interacts with the TLR2/MyD88 signaling pathway, thus resulting in the decreased expression of SNAP-23 in LPS stimulated microglial cells. The results from Piotroska et al. [57] are in line with the results of Hepp et al. [58], who reported that SNAP-23 replaces SNAP-25 in microglia and oligodendrocytes. The effects of BoNT/A on SNAP-23 seem to contrast with the natural molecular targets of BoNT/A in neuronal cells (i.e., SNAP-25); however, when analyzing the structural features of SNAP-25, and its non-neuronal SNAP-23 isoforms, which include the murine mSNAP-23 and human hSNAP-23, Vaydianathan et al. [59] found that BoNT/A was effectively able to cleave the non-neuronal mSNAP-23, but not hSNAP-23. Additionally, BoNT/E was more efficient than BoNT/A in cleaving mSNAP-23. Notably, if BoNT/A was only able to block microglial mSNAP-23, this finding would pose an important limitation, in that it would highlight the difficulty in translating these results to a therapeutic setting for humans, from a pharmacological perspective (i.e., regarding the possibility of interacting BoNT/A with microglia as an alternative method to treat chronic pain). Further research is needed to clarify these points.

In a rat model involving a spinal cord injury (SCI), Yu et al. [60] observed that the combined application of BoNT/A (injection of two doses of 1.25 U of Botox^®^ around the SCI site and forelimb muscle) and minocycline, an antibiotic agent that also has anti-inflammatory properties, synergistically reduced neuropathic pain and apoptosis by inactivating the glial cells. In further detail, the authors found that the combination of BoNT/A and minocycline promotes the expression of the SIRT1 cell signaling pathway, inactivates the NF-κB, P53, and PI3K/AKT signaling pathways, and attenuates an inflammatory response and oxidative stress. These combined effects greatly enhance the therapeutic effect of the two drugs.

Feng et al. [61] showed that a single intraplantar, or the intrathecal, pre-administration of BoNT/A (0.5–1 U/kg of Botox^®^), in a rat that was subjected to a partial sciatic nerve ligation (PSNL) pain model, significantly prevented PSNL-induced allodynia and thermal hyperalgesia, together with a reduced upregulation of pro-inflammatory cytokines in the spinal cord, dorsal horn, and dorsal root ganglions (DRGs). In order to determine the direct effect of BoNT/A on microglia and/or astrocytes, the authors also performed an in vitro experiment on the LPS-activated glial cells that were treated with BoNT/A (1–2 U/mL of Botox^®^). They found that BoNT/A significantly inhibited the activation of LPS-activated microglia and reduced the release of TNF-α, IL-6, Il-1β, iNOS, and MIP-1α, without any effect on the astrocytes’ activation. This latter result appears to be in contrast with findings that detect the cl-SNAP25 immunoreaction after treatment with BoNT/A, both in LPS-activated astrocytes [56] and in spinal astrocytes, either in CCI pain models [18,62] or in spinal cord injury models [63]. Interestingly, Finocchiaro et al. [64] reported a strong reduction in the activation of spinal astrocytes, a study which also involved CCI mice that were treated with BoNT/B (intraplantar injection of 7.5 pg/mouse of 150 KDa of purified BoNT/B). Conversely, no difference in the expression of resting and activated microglia were observed in CCI mice treated with BoNT/B. The discrepancies observed in the different BoNTs serotypes may depend on the different targets of the toxins, namely, SNAP-25 for BoNT/A and VAMP-2 for BoNT/B, and the different expressions of these targets in neuronal and non-neuronal cells.

The possible interaction between BoNTs and glial cells was not only analyzed in the context of neuropathic pain, but also in the context of inflammatory pain. In a chronic inflammatory pain model, which involved an intraarticular injection of a solution of complete Freund’s adjuvant (CFA) into the ankle joint cavity of the left leg of a rat, Shi et al. [65] observed that an intraarticular injection of BoNT/A (5–10 U/Kg of Botox^®^ into ankle articular cavity) reduced CFA-induced pain-related behaviors in a dose dependent manner. Similar behavioral effects were achieved by blocking the activation of spinal microglia and reducing TNF-α. Furthermore, the authors found that the effect of BoNT/A on spinal microglial activation was associated with the inhibition of spinal microglial P2X4R–P38MAPK intracellular signaling pathways. Similarly, in a model that involved antigen-induced arthritis of the temporomandibular joint (TMJ) in rats, which was induced by injecting and emulsifying CFA and methylated serum albumin, Munoz-Lora et al. [66,67] found that the intra-TMJ injection of BoNT/A (7 U/Kg of Botox^®^, or 14 U/Kg of Dysport^®^) was able to reduce the P2X7/Cathepsin-S/Fractalkine microglia-activated pathway in the trigeminal subnucleus caudalis. Moreover, BoNT/A also reduced the protein level of IL-1β and TNF-α.

In all the studies presented thus far, the exact mechanism by which the peripherally injected BoNT/A reaches the spinal cord, where it may block both neuronal synaptic release and spinal glial activation, is not yet completely understood. As has been suggested in many studies that have used animal models [68], a direct central effect of the peripheral administration of BoNT/A is conceivable as a consequence of its retrograde transport along the axons of sensory neurons and its subsequent transcytosis to neuronal and non-neuronal spinal cells, where it can block both the release of neurotransmitters and the activation of spinal glia cells. It should be noted that, although the retrograde transport of the toxin can be evoked as a mechanism by which the peripheral toxin can reach the spinal cord in animal models, for obvious reasons, it is desirable that this does not happen in humans. The retrograde transport of the toxin from the peripheral injection site, which is an uncontrollable event, is an undesirable adverse effect, and in practical medicine, every effort is aimed at ensuring that this event does not occur. In light of this, it is unthinkable to consider the possibility of translating this type of mechanism, which concerns the toxin–glial cell interaction, for use in a clinical setting. Nonetheless, this problem could be circumvented by synthesizing new chimera toxins, which, if they are successfully designed to recognize specific receptors, may selectively target glial cells. In recent years, the development of engineered toxins has become the subject of intense research in the field of botulinum toxins, and it is desirable that this continues further [69,70,71].

## 5. Interaction between BoNTs and Myelin Forming Cells

Oligodendrocytes and SCs are the myelin-forming cells within CNS and PNS, respectively. A recent paper on the therapeutic potential of BoNT/A in counteracting paralysis and neuropathic pain, which used a model involving spinal cord injuries (SCI) in mice, analyzed the possible interaction between BoNT/A and oligodendrocytes [63]. Oligodendrocytes are highly susceptible to spinal cord damage that is induced by traumatic injury, and they easily undergo apoptosis as consequence of SCI [72]. Accordingly, Vacca et al. [63] observed a massive expression of the apoptotic marker Caspase 3, which is partially co-localized with oligodendrocytes, at the epicenter of the spinal impact. An intrathecal injection of BoNT/A (15 pg/mouse of 150 KDa purified BoNT/A in spinal cord) significantly reduced Caspase 3 expression, thus indicating protection against apoptotic processes. In parallel to this, the spared tissue was also abnormally myelinated, with a consequent reduction in axonal conduction. As a reactive response, to induce the healing of the injured spinal cord, the surviving oligodendrocytes produce a myelin basic protein (MBP) [73]. Vacca et al. [63] found an increased expression of MBP in saline-treated SCI mice, whereas in BoNT/A-treated SCI mice, the MBP was approximatively at the same level as the naïve mice, thus indicating that there was support for the reduced reactivity of oligodendrocytes due to a minor degree of degeneration.

In the last decade, convincing evidence that supports the interactive effect of BoNT/A on SCs proliferation has also been obtained, mainly in CCI neuropathic mice. Marinelli et al. [74] showed that the intraplantar injection of BoNT/A (15 pg of 150 KDa of purified BoNT/A/paw) in CCI mice prompts the functional recovery of injured hindlimbs, accelerates regenerative processes, and enhances the expression of proliferative cells in sciatic nerve tissue. In a subsequent study that was performed in naïve mice, the mice underwent an intraplantar injection of BoNT/A (15 pg of 150 KDa of purified BoNT/A/paw). Marinelli et al. [18] detected cl-SNAP-25 along the nociceptive pathway, which started at the hind paw skin and ended at the spinal cord. In particular, in the sciatic nerve, cl-SNAP-25 co-localized with a glial fibrillar acidic protein, which is a protein expressed in unmyelinated fibers and in de-differentiated SCs after injury [75]. To determine if BoNT/A may directly interact with the proliferative state of SCs, Marinelli et al. [18] performed an in vitro experiment and found that the ACh released from SCs was reduced using BoNT/A, thus suggesting that BoNT/A may not only cleave SNAP-25 in neuronal cells, but also in SCs. In accordance with this finding, the presence of SNAP-25 in SCs has been demonstrated by Barden et al. [76] who found a receptor cluster located in SCs, which was composed by SNAP-25, together with other SNARE proteins, N-type calcium channels, and P2X receptors. The ACh release from SCs, which was reduced by BoNT/A, may affect the proliferation of SCs. With this in mind, it should be noted that low levels of ACh in the environment that surrounds the SCs, stimulates SC proliferation, whereas high levels of ACh arrests SC proliferation [77,78]. After CCI in vivo, the ACh that is released from the SCs, and from axons undergoing myelination, may be blocked by BoNT/A, thus causing a consequent reduction of ACh levels in the microenvironment that surrounds the regenerating nerve. This finding was confirmed by Cobianchi et al. [79], who observed that a single intranerve injection of BoNT/A (15 pg of 150 KDa of purified BoNT/A/paw) in mice stimulates the regeneration of myelin fibers and the speed of axonal elongation after an injury to the peripheral nerve; this occurs due to the activation and proliferation of SCs. Similar results were obtained by Seo et al. [80], who observed an increased proliferation of SCs after an intranerve injection of BoNT/A (7 U/Kg of Botox^®^) in the crushed sciatic nerve of rats. It is notable that, in contrast to what was observed with BoNT/A, BoNT/B reduces the proliferation of SCs; this could be explained by considering the different targets of the two BoNTs [64].

## 6. Interactions between BoNTs and Other Glial Cells

Regarding the SGCs, Da Silva et al. [81] demonstrated that trigeminal SGCs express the docking proteins, SNAP-25 and SNAP-23. Moreover, authors observed that a pre-treatment with different concentrations of BoNT/A (medium containing 0.1, 1, 10, and 100 pM of Botox^®^) inhibited, in a concentration-dependent manner, the ionomycin-evoked release of glutamate from cultured SGCs, thus indicating a possible direct interaction between BoNT/A and SCGs through the cleavage of SNAP-25 and/or SNAP-23, which would inhibit the vesicular release of glutamate. With this in mind, SGCs appear as potential targets for future therapeutic options in the treatment of chronic pain [82]. Another in vitro study on cultured trigeminal ganglions (TG) demonstrated a direct interaction between BoNT/A and sensory mechanisms [83]. Accordingly, the authors observed the expression of both SNAP-25 and BoNTs receptor SV2-A in TG. The expression of SNAP-25 ed SV2-A was also observed in sphenopalatine ganglions that were isolated from rats and human autopsies [84]. Interestingly, a mixed expression was observed between rats and humans. In rats, SV2-A was expressed in SGCs and SNAP-25 in neurons, whereas the opposite was observed in humans. The significance of the differential expressions between SNAP25 and SV2-A remain unclear. Among the other glial cells, namely, ependymal, radial, and enteric glial cells, no evidence of a possible interaction between BoNTs and these cells has been presented until now. 

## 7. Summary

In this review, the most recent evidence that highlight a possible interaction between BoNTs, mainly BoNT/A, and glial cells was presented and discussed. Two main conclusions can be drawn from the cited articles, which might be considered as the strengths of this review. First, it is confirmed that BoNT/A is able to block the release of neuroactive substances, not only from neurons, but also from glial cells. This action can contribute, for example, to the analgesic activity of BoNT/A for chronic pain, during which the activation of microglia and astrocytes, secondary to the onset of painful phenomenology, contributes to the amplification and maintenance of chronic pain itself. Second, the interaction between BoNT/A and glial cells is not only confined to microglia and astrocytes. In fact, although there are still only a few studies on this matter, BoNT/A can interact with the glial cells that are responsible for the reconstruction of the myelin sheath of neural axons, such as SCs and oligodendrocytes, and thus the gradual restoration of the myelin function may occur even after it has been subjected to traumatic injury.

Although the strengths of this review are undeniable, at the same time, it has some weaknesses. First of all, unfortunately, there are still few publications on this topic. This is why the explanation of the results presented in this paper can sometimes appear incomplete and controversial. Secondly, since the effects of BoNTs on glial cells are very complex, and due to the diversity of the models used and the diversity of the glial cells themselves, it is impossible, at this stage, to formulate a unifying mechanism of action. Despite these weaknesses, this review still aims to establish a starting point for stimulating further research on the interaction between BoNTs and glial cells in order to elucidate the mechanism of action, which is still far from being defined.

## 8. Concluding Remarks and Future Perspectives

Although it still seems utopian to think of using BoNTs as agents that are capable of blocking the activation and release of pro-inflammatory substances from glial cells in humans, the door is open in terms of conducting more research on this topic. Certainly, this constitutes a field of investigation that is worth being continuously explored in order to clarify questions, mainly regarding the mechanism of action of BoNTs, which are still unsolved; for example, a fundamental contribution should come from a definitive clarification of the hypothesized retrograde transport of BoNTs. In particular, it would be useful to precisely clarify the following questions:Are all the effects of peripheral BoNTs on central spinal glial cells solely due to direct interactions that occur after the retrograde transport of BoNTs?How should this retrograde transport be controlled so that it does not cause adverse effects after the BoNTs are injected?Which receptors on the membrane of the glial cell allow BoNTs to enter the glial cell and exert their proteolytic effect?Is the mechanism of action of BoNTs inside the glial cells, as has been widely demonstrated at the neuronal level? Are there specific internal glial cell targets involved?

These are just some of the questions that future studies on this topic should answer. On a final note, it is worth considering that, once again, this fascinating molecule, the botulinum toxin, never ceases to amaze with its multiple applications. Given that the therapeutic use of this molecule began in the 70s, with the pioneering work of Dr. Alan B. Scott and colleagues, who applied BoNT/A in the ophthalmological field, there are no other molecules in nature that have shown such a variety of effects and therapeutic uses, both registered and non-registered. Surely, in the future, BoNTs will continue to surprise.

## Figures and Tables

**Figure 1 toxins-14-00704-f001:**
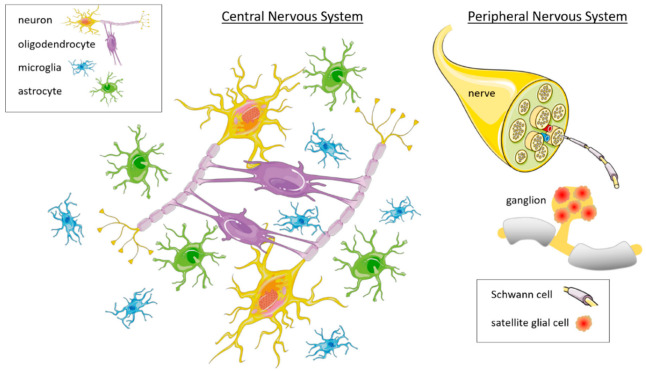
Diagram of the main types of glial cells in the central and peripheral nervous system. In the CNS, glial cells include astrocytes, oligodendrocytes, and microglia, whereas in the PNS, the glial cells include SCs and SGCs. Other glial cells such as ependymal, radial, and enteric glial cells are not depicted in this diagram. Parts of this figure were composed using pictures from Servier Medical Art (accessed on 18 July 2022 from http://smart.servier.com), a free service provided by Les Laboratoires Servier (accessed on 18 July 2022 from http://www.servier.com). Servier Medical Art is licensed under a Creative Commons Attribution 3.0 Unported License (accessed on 18 July 2022 from https://creativecommons.org/licenses/by/3.0/).

## Data Availability

Not applicable.

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
