# Peer review of "Botulinum Neurotoxins beyond Neurons: Interplay with Glial Cells"

_toxins, 2022, doi:10.3390/toxins14100704_

Round 1
Reviewer 1 Report
This was interesting review, but extensive revision should be done for the publication level.
Major flaw: The similarity score was too high. Authors should not do just "copy and paste" of published articles.
Minor flaws
1. The abstract was written for general information of BTX. As the objective of this study was the effect of BTX on glial cells, the summary of update on this issue should be mentioned in the abstract.
2. In key contribution section, the examples of pathologies should be mentioned.
3. "in vitro" and "et al" should be italic.
4. Please check typo errors. For example, "sinergistically "
5. The content of reference [58] by Yu was strange. The mechanism of BTX and minocycline had been explained as promoting the expression of SIRT1 cell signalling pathway and inactivating NF-κB, P53, and PI3K/AKT signallling pathway, attenuating inflammatory response and oxidative stress. Minocycline is a tetracycline antibiotic medication, not an anti-inflammatory drug. BTX may inhibit exocytosis of intracellular vesicle. It can influence on some autocrine agents.
6. Please specify the name of pro-inflammatory cytokines in line 227.
7. "is not yet completely cleared up " can be understood for general conversation, but not scientific expression.
8. The cartoon for summary of section 3, 4, and 5 should be added. There were only scattered findings from several authors.
Author Response
Reply to Reviewer #1
This was interesting review, but extensive revision should be done for the publication level
RE: I would like to thank the reviewer for its revision of the paper and positive evaluation of the review.
Major flaw: The similarity score was too high. Authors should not do just "copy and paste" of published articles
RE: The similarity score has been reduced and many paragraphs have been rephrased. This was done also based on requirements from Editorial Office.
Minor flaws
- The abstract was written for general information of BTX. As the objective of this study was the effect of BTX on glial cells, the summary of update on this issue should be mentioned in the abstract.
RE: abstract have been completely rewritten
- In key contribution section, the examples of pathologies should be mentioned.
RE: key contribution has been rewritten
- "in vitro" and "et al" should be italic.”
RE: “in vitro” has been written in italic; at difference, “et al” was not changed to italic since it does not seem Toxins' writing style.
- Please check typo errors. For example, "sinergistically"
RE: I apologize about many typos errors present in the original version. In the revised version, all typos errors have been corrected.
- The content of reference [58] by Yu was strange. The mechanism of BTX and minocycline had been explained as promoting the expression of SIRT1 cell signaling pathway and inactivating NF-κB, P53, and PI3K/AKT signaling pathway, attenuating inflammatory response and oxidative stress. Minocycline is a tetracycline antibiotic medication, not an anti-inflammatory drug. BTX may inhibit exocytosis of intracellular vesicle. It can influence on some autocrine agents.
RE: Thank you for this comment but I must disagree with the reviewer. I don’t think that the work of Yu et al [58] was strange. In fact, it is true that minocycline is an antibiotic, but on the same time it may exert anti-inflammatory (see Giuliani et al. (2005) J Leukoc Biol 78, 135), anti-apoptotic (see Popovich et al. (2002) Ann Neurol 51, 215), and neuroprotective effects (see Song et al. (2004) Neuroreport 15, 2181). These properties are independent of its antimicrobial activity. The non-antimicrobial properties have been attributed to inhibition of the activity of caspase-1 and -3, and enhancement of BcL-2 (Chen et al. (2000) Nat Med 6, 797; Domerq & Matute (2004) Trends Pharmacol Sci 25, 609)
- Please specify the name of pro-inflammatory cytokines in line 227.
RE: name of pro-inflammatory cytokines has been provided
- "is not yet completely cleared up" can be understood for general conversation, but not scientific expression.
RE: the expression has been rephrased.
- The cartoon for summary of section 3, 4, and 5 should be added. There were only scattered findings from several authors.
RE: Since the effects of botulinum toxin on glial cells are very complex, and due to diversity of model used and diversity of glial cells itself, it is impossible at this stage to draw a summarizing cartoon. The purpose of the review was precisely to set a starting point to stimulate further research on the effects of botulinum toxins on glial cells in order to clarify the mechanism of action, which is still far from being defined.
Reviewer 2 Report
This was an interesting paper to read and well structured. I only have a few suggestions:
Line 34 et seq, line 296 BoNT are blockers of synaptic vesicle fusion, not inhibitors, as stated in lines 304 & 337
Lines 42-45 Citation needed
Lines 98-99 Citation needed
Lines 137-139 Citation needed
Citation 49 is incomplete
Lines 169-178 Citations needed
Line 180 In reviewing the Gui et al (2020) study, the authors should comment on the very high levels of BoNT used. At 20 units/Kg, this would represent a dose of 1500 units for a 75Kg adult, which is exceptionally high (actually, 15 vials at 100u per vial). The use and value of such an animal model therefore has to be questioned. The same comment applies to the study cited in [53].
Lines 258-265 The authors propose a mechanism of retrograde transport of BoNT. They should therefore comment on whether effects are seen in humans that would support this mechanism, after patients are treated with BoNT. Especially in the case of high dose treatment, for example in spasticity.
Author Response
Reply to Reviewer #2
This was an interesting paper to read and well structured. I only have a few suggestions:
RE: I would like to thank reviewer for its positive consideration of the review
Line 34 et seq, line 296: BoNT are blockers of synaptic vesicle fusion, not inhibitors, as stated in lines 304 & 337
RE: Thank you for this observation. Text has been corrected.
Lines 42-45 Citation needed
Lines 98-99 Citation needed
Lines 137-139 Citation needed
Citation 49 is incomplete
Lines 169-178 Citations needed
RE: All the required citations have been added.
Line 180: In reviewing the Gui et al (2020) study, the authors should comment on the very high levels of BoNT used. At 20 units/Kg, this would represent a dose of 1500 units for a 75Kg adult, which is exceptionally high (actually, 15 vials at 100u per vial). The use and value of such an animal model therefore has to be questioned. The same comment applies to the study cited in [53].
RE: I thank the reviewer for its observation. As suggested, a comment on the high levels of BoNT used in the cited studies has been added.
Lines 258-265: The authors propose a mechanism of retrograde transport of BoNT. They should therefore comment on whether effects are seen in humans that would support this mechanism, after patients are treated with BoNT. Especially in the case of high dose treatment, for example in spasticity.
RE: Regarding the possible retrograde transport of the toxin, I am aware of the fact that this should be avoided as much as possible in humans. Nevertheless, there are numerous works that highlight the fact that many of the effects seen in animal models can only be explained by evoking a retrograde transport of the toxin from the peripheral injection site. The clinical translation of data from animal models to humans will only be possible by bypassing this mechanism and studying the possibility of having toxins specifically engineered to be injected centrally into targeted neuronal districts. A comment on this has been added in the revised version.
Round 2
Reviewer 1 Report
Authors improved the manuscript compared to previous version. However, I feel that there may be not enough publications to be reviewed for thus topic. That's the reason why the explanation of the authors is incomplete and controversial.
1. Authors stated that minocycline has anti-inflammatory property. It may be. However, the mechanism is still unclear. Authors stated that activation of SIRT1 is one of its anti-inflammatory mechanism. Activation of SIRT1 can improve cellular survival and has been observed after the administration of phenolic compound. But the activation of SIRT1 is not directly linked to anti-inflammation. NF-kB is a key signalling molecule for the inflammatory reaction. However, it was unclear how minocycline could inhibit NF-kB. P53 also has many functions in the cellular activity. Why it should be linked anti-inflammatory only?
2. Authors refused the adding of cartoon. The reason was "Since the effects of botulinum toxin on glial cells are very complex, and due to diversity of model used and diversity of glial cells itself, it is impossible at this stage to draw a summarizing cartoon. The purpose of the review was precisely to set a starting point to stimulate further research on the effects of botulinum toxins on glial cells in order to clarify the mechanism of action, which is still far from being defined." As authors stated that it is impossible to summarize current status on this topic. Then, it is clear that this topic is too much preliminary for review.
Author Response
Reply to Reviewer #1
1. Authors stated that minocycline has anti-inflammatory property. It may be. However, the mechanism is still unclear. Authors stated that activation of SIRT1 is one of its anti-inflammatory mechanism.…….Why it should be linked anti-inflammatory only?
RE: I honestly don't understand the reviewer's aversion to this part of the review which reports an article on the synergistic effect of the toxin along with minocycline. I am not an expert on minocycline and I have only reported the explanation of the authors of cited article. Whether it is right or wrong I have neither the competence nor the presumption to affirm it. About the fact that minocycline can act as an inflammatory agent there are a number of indications for this and I have cited some articles in favor. I don't think there is still doubt about this.
2. Authors refused the adding of cartoon.…….. As authors stated that it is impossible to summarize current status on this topic. Then, it is clear that this topic is too much preliminary for review.
RE: Evidently the reviewer did not understand that the purpose of the review is to stimulate further studies on a topic towards which future research efforts, especially basic ones, should focus more. I agree with the reviewer that the preliminary nature of these data, which however are starting to be of some relevance, does not allow to define a clear mechanism of action, nevertheless I do not believe that such a topic cannot be treated in a review anyway, even if the data may appear preliminary.